# Gradual Stress-Relaxation of Hydrogel Regulates Cell Spreading

**DOI:** 10.3390/ijms23095170

**Published:** 2022-05-05

**Authors:** Wenting Yu, Wenxu Sun, Huiyan Chen, Juan Wang, Bin Xue, Yi Cao

**Affiliations:** 1Oujiang Laboratory (Zhejiang Lab for Regenerative Medicine, Vision and Brain Health), Wenzhou Institute, University of Chinese Academy of Sciences, Wenzhou 325001, China; yuwenting@ucas.ac.cn; 2Jinan Microecological Biomedicine Shandong Laboratory, Shounuo City Light West Block, Qingdao Road 3716#, Huaiyin District, Jinan 250117, China; 3School of Sciences, Nantong University, Nantong 226001, China; sunwenxu@ntu.edu.cn; 4Collaborative Innovation Center of Advanced Microstructures, National Laboratory of Solid State Microstructure, Key Laboratory of Intelligent Optical Sensing and Manipulation, Ministry of Education, Department of Physics, Nanjing University, Nanjing 210093, China; dg20220003@smail.nju.edu.cn (H.C.); wangjuannm@163.com (J.W.)

**Keywords:** hybrid network hydrogel, stress-relaxation gradient, peptide-metal ion coordination, cell spreading, mechanical property

## Abstract

There is growing evidence that the mechanical properties of extracellular matrices (ECMs), including elasticity and stress-relaxation, greatly influence the function and form of the residing cells. However, the effects of elasticity and stress-relaxation are often correlated, making the study of the effect of stress-relaxation on cellular behaviors difficult. Here, we designed a hybrid network hydrogel with a controllable stress-relaxation gradient and a constant elasticity. The hydrogel is crosslinked by covalent bonds and dynamic peptide-metal ion coordination interactions. The stress-relaxation gradient is controlled by spatially controlling the coordination and covalent crosslinker ratios. The different parts of the hydrogel exhibit distinct stress-relaxation amplitudes but the have same stress-relaxation timescale. Based on this hydrogel, we investigate the influence of hydrogel stress-relaxation on cell spreading. Our results show that the spreading of cells is suppressed at an increasing stress-relaxation amplitude with a fixed elasticity and stress-relaxation timescale. Our study provides a universal route to tune the stress-relaxation of hydrogels without changing their components and elasticity, which may be valuable for systematic investigations of the stress-relaxation gradient in cell cultures and organoid constructions.

## 1. Introduction

Cell behaviors, such as spreading [1], migration [1,2,3,4], proliferation [5,6,7], and differentiation [8,9], are often regulated by the mechanical properties of ECMs. ECMs exhibit complex mechanical behaviors, including viscoelasticity or elasticity, which are determined by their components [10]. In vivo, ECMs are heterogeneous mixtures whose component spatial distribution can lead to mechanical gradients [11,12], including the elasticity gradient and viscoelasticity gradient. The impacts of the elasticity gradient on cell behaviors have been widely studied in recent decades. For example, in early mouse limb buds, the mesodermal stiffness gradient can prompt cells to migrate collectively, which is important in bud development [13]. The stiffness gradient in Xenopus laevis placodal tissues can lead to the migration of neural crests, which is important in nerve tube development [14]. The stiffness gradient in the Xenopus laevis brain [15] can induce axon spreading toward soft areas. 

Furthermore, ECM viscoelasticity can also regulate cell behaviors, such as cell spreading [16,17,18], migration [19,20,21,22], proliferation [17], and differentiation [23,24,25]. In vivo, ECMs usually exhibit gradient stress-relaxation under constant deformations [26,27]. The components of natural ECMs are complex and uncontrollable, making it difficult to be used as a model system for studying the effects of stress-relaxation on cell behaviors [12,28]. By taking advantage of hydrogels and mimicking the mechanical properties of ECMs [29,30,31,32,33,34,35,36,37,38,39], researchers have developed hydrogels with controllable stress-relaxation to investigate the impacts of the timescale of stress-relaxations on cell behaviors, including cell spreading and differentiation [17,40,41]. However, hydrogels with controllable stress-relaxation amplitudes and the effects on cell behaviors have rarely been explored. Generally, the amplitude of stress-relaxation depends on the ratio of noncovalent and covalent crosslinking (*ε*) in hydrogels, while the timescale of stress-relaxation depends on the intrinsic kinetics (*τ**_c_*) of dynamic crosslinking [40,42]. The elasticity of hydrogels correlated to the overall density (*η*) of different crosslinking [43,44,45,46]. In order to investigate the effects of stress-relaxation amplitudes on cell behaviors, the *ε*, *τ*, and *η* need to be perfectly balanced, which remains a challenge.

To this end, we designed a hydrogel with gradient stress-relaxation amplitudes while the stress-relaxation timescale and elasticity remained consistent (denoted as SRG hydrogel hereafter). By altering the density of metal coordination and covalent crosslinkers, the varied stress-relaxation amplitude was achieved, and the spatial distribution of the stress-relaxation with different amplitudes was systematically studied. Based on this hydrogel, we studied the effect of stress-relaxation on cell adhesion and spreading. Our work revealed that cell spreading is suppressed with increasing stress-relaxation amplitudes at a fixed elasticity and stress-relaxation timescale. Our results provide a general approach for tuning the stress-relaxation of hydrogels without changing their components and elasticity. The design principle presented in this work may guide future efforts for systematic investigation of the stress-relaxation gradient on cell cultures and organoid constructions. 

## 2. Results and Discussion

### 2.1. Designing Hydrogels with a Stress-Relaxation Gradient

To construct the SRG hydrogel, we introduced both dynamic and permanent crosslinkers into a hybrid network hydrogel [47,48,49,50,51]. The SRG hydrogel was formed by acrylamide, four-armed PEG-Aclt, histidine-rich peptide (acrylate-PH_6_), and Zn^2+^. Acrylate-terminated multiarmed PEG was introduced into the hydrogel network to act as covalent crosslinkers. The complex formed by the coordination of PH_6_ peptide and Zn^2+^ (PH_6_-Zn^2+^ complex) was used to act as the dynamic crosslinker [52,53,54,55,56]. In a typical hydrogel preparation, precursor solutions at different PEG concentrations were added into the mold layer by layer to achieve a gradient of covalent crosslinkers via molecular diffusion, and the pre-gel was formed by copolymerization of acrylamide, four-armed PEG-Aclt, and acrylate-PH_6_ peptide (Figure 1A left). Then, the band-shaped hydrogel was immersed in ZnCl_2_ solution step by step to obtain a gradient of dynamic crosslinkers (Figure 1A right). As a result, the coordination density in the hydrogel varied with the diffusion of Zn^2+^ from the bottom to the top of the band-shaped hydrogel. Since Young’s modulus of the hydrogel is correlated to the crosslinking density according to the swollen elastic rubber theory [43,44,45,46], the Young’s modulus of different locations of the hydrogel could remain consistent at a fixed total density of the covalent and noncovalent crosslinking. In contrast, the amplitudes of the stress-relaxation for different locations on the hydrogel would vary along with the gradient of the PH_6_-Zn^2+^ complex because the stress-relaxation property is correlated with the dissociation of dynamic crosslinking. Herein, we fabricated an SRG hydrogel with a consistent Young’s modulus and gradient stress-relaxation properties by altering the coordination and covalent crosslinker densities. As shown in Figure 1B,C, the ratio of PH_6_-Zn^2+^ crosslinkers (crosslinking density of PH_6_-Zn^2+^ complex/total density of all the crosslinking) varied from 0% to 68% along the long axis of the SRG hydrogel, and different locations of the hydrogel were named using the corresponding PH_6_-Zn^2+^ crosslinker ratio. Figure 1D shows the porous microstructures of three typical locations of a band-shaped SRG hydrogel (0%, 34%, and 68%), and no obvious difference was observed, further confirming the homogeneity of the hydrogel network.

### 2.2. Mechanical Properties of the SRG Hydrogels

The mechanical properties of the SRG hydrogels were quantitatively evaluated using mechanical compression tests. Typical stress-strain curves of different locations on the hydrogels with and without coordination are shown in Figure 2A and Appendix A. As summarized in Figure 2B, the Young’s modulus of different locations from the SRG hydrogel remained constant (~140 kPa), while those of the pre-gel in the absence of Zn^2+^ increased from 50 kPa to 140 kPa along the axis. The fracture stress and toughness of the hydrogel with a stress-relaxation gradient increased from 139.35 kPa and 41.54 kJ m^−3^ as the PH_6_-Zn^2+^ crosslinker ratio increased from 0% to 68% (Figure 2C). Various PH_6_-Zn^2+^ densities in the separate hydrogels were achieved by immersion in Zn^2+^ solutions for different lengths of time. Moreover, the mechanical hysteresis of the hydrogels was also investigated via compression-relaxation cycles. As indicated in Figure 2D,E, the dissipated energy of different locations of the hydrogel increased from 1.57 kJ m^−3^ to 5.19 kJ m^−3^ as the coordination crosslinker ratio increased. Since the dissipated energy in the loading-unloading cycles is linked to the rupture of the dynamic crosslinkers in the hybrid network hydrogel, the amplitude of the energy dissipation increased with the scale-up of the PH_6_-Zn^2+^ density in the hydrogel. The maximum stresses remained constant in compression-relaxation cycles because the stress mainly contributed to the crosslinking density. It is worth mentioning that the swelling ratios and water contents of the different locations of the hydrogels were almost the same, further indicating the homogeneity of the hydrogel network (Appendix A).

More importantly, SRG hydrogels exhibited gradient stress-relaxation behaviors, as expected. As evaluated by the stress-relaxation tests, the stress-relaxation amplitudes of different locations of the hydrogel varied with the PH_6_-Zn^2+^ crosslinker ratios (Figure 2F), while the characteristic relaxation timescale remained almost unchanged (Figure 2G). The stress-relaxation amplitudes increased from 12.88 kPa to 36.56 kPa, and the characteristic relaxation times (*τ*) remained at 3 min, as summarized in Figure 2H. All these results indicated that a hydrogel with consistent elasticity and gradient stress-relaxation amplitude was successfully fabricated through the layer-by-layer building of pre-gels and the gradient distribution of peptide-ion coordination.

### 2.3. Cell Spreading on SRG Hydrogels

Based on the SRG hydrogel, we investigated the effects of stress-relaxation gradients on cell spreading, including actin skeleton and focal adhesion. The biocompatibility of the hydrogels was first evaluated using the Cell Counting Kit-8 (CCK-8) with L929, HeLa, and MCFs cells as the model cells. As shown in Appendix A, the cell viabilities remained more than 80% for all the cells compared to those of the control medium, indicating the outstanding biocompatibility of the hydrogels. 

To observe the difference in cell behaviors affected by the stress-relaxation gradient, we selected five typical locations along the long axis of the SRG hydrogel (0%, 17%, 34%, 51%, and 68%). After being seeded on the hydrogel, the L929 cells were imaged every 3 h for the next 15 h. No obvious difference in cell densities was found in the different locations of the SRG hydrogels due to the fact that the modulus remained consistent. However, distinct cell spreading was observed in the different locations of SRG hydrogels. Generally, the fusiform cell percentages increased over time (Figure 3A and Appendix A), while fewer fusiform cells were observed with the increase in the PH_6_-Zn^2+^ ratio (Figure 3B,C). As indicated in Figure 3D and Appendix A, the cell area decreased with the increase in the PH_6_-Zn^2+^ ratio after being cultured for the same time. Meanwhile, the average spreading rate of the L929 cells decreased with the increase in the PH_6_-Zn^2+^ ratio (Figure 3E). The spreading rate in 15 h decreased from 4.4 to 1.45 μm^2^ h^−1^ as the PH_6_-Zn^2+^ ratio increased from 0% to 68%. However, the circularity of L929 cells (cell width/cell length) obviously increased with the increase in the PH_6_-Zn^2+^ ratio. These results suggested that the cell spreading behaviors were significantly suppressed by the enhanced stress-relaxation amplitude at a fixed elasticity and stress-relaxation timescale. 

Next, the spreading behaviors of HeLa cells and MCFs on the SRG hydrogels were also investigated. The spreading of both kinds of cells exhibited similar trends as that of L929 cells on different locations of SRG hydrogels (Appendix A). As summarized in Figure 3F–H, Appendix A, the HeLa and MCFs exhibited decreased spreading areas and rates as the PH_6_-Zn^2+^ crosslinker ratio increased after being cultured for the same time. Moreover, the circularity monotonically increased with the increase in PH_6_-Zn^2+^ ratios. All these results indicated that the suppression of cell spreading from the increasing stress-relaxation amplitude was general for various kinds of cell lines.

### 2.4. Hypothesis of Mechanisms for Cell Spreading Regulation with the Gradual Stress-Relaxation

Our results show that hydrogels with stress-relaxation could suppress cell spreading. However, it remains unclear how this suppression was achieved. For cells cultured in ECM, ECM coupled clutches inhibit the retrograde flow of actin and drive cell spreading. Meanwhile, actin requires persistent resistance to cellular traction forces for further assembly and allows cell spreading (Figure 4A) [57,58]. In contrast, the dynamic coordination crosslinkers in hydrogels would allow the remodeling of hydrogel networks under deformations, which may relax the resistance to cellular traction forces over time. The relaxed resistance to cellular traction forces resulted in the suppression of cell spreading. 

For the 0% location on SRG hydrogels, the amplitude of stress-relaxation was mild due to the low density of PH_6_-Zn^2+^ crosslinkers. Hydrogel resistance to cellular traction forces almost does not relax over time, leading to fast cell spreading (Figure 4B). For the 34% location on SRG hydrogels, the enhanced amplitude of stress-relaxation resulted in the moderate suppression of cell spreading (Figure 4C). In contrast, the significantly increased density of the PH_6_-Zn^2+^ crosslinkers for the 68% location on SRG hydrogels contributed to the much-enhanced amplitude of stress-relaxation, leading to the major suppression of cell spreading (Figure 4D). This hypothesis can be further confirmed by the cell spreading on hydrogels without stress-relaxation gradients. As shown in Appendix A, the HeLa cells were cultured on covalently crosslinked homogeneous hydrogels with consistent elasticity. Compared to those cells on the SRG hydrogels, no obvious difference in cell spreading was observed, and the spreading rate was much faster. This confirmed that hydrogel stress-relaxation, parallel to stiffness, influences cell spreading by relaxing the resistance to cellular traction forces. 

Note that the stress-relaxation of hydrogels affecting cell spreading by relaxing the resistance to cellular traction forces is consistent with some of the previous reports. For example, Mandal et al. [59] reported that normal hepatocytes cultured on elastic substrates spread faster and larger than those cultured on viscoelastic substrates. Charrier et al. [60] found that HASM and 22Rv1 cells cultured on a pure elastic matrix spread larger than those cultured on a stress-relaxation matrix. However, Fabry and coworkers reported that fibroblasts elongate, migrate and proliferate better in hydrogels that display a higher stress relaxation amplitude [16]. For cells in 3D culture, the elongation, migration, and proliferation of cells are limited by the hydrogel network. The high stress-relaxation amplitude reduced the limitation of penetrating and remodeling the matrix for cells. In contrast, the actin requires persistent resistance to cellular traction forces for further assembly and thus enhances the cell spreading for cells spreading on hydrogels in this work. The increased stress-relaxation amplitude in hydrogels would increase the relaxation of the resistance to cellular traction forces over time, leading to the suppression of cell spreading.

A unique advantage of the hydrogels reported in this work is that the SRG hydrogel shows tunable stress-relaxation behaviors without changing its components and elasticity [41,43]. By changing the coordinate crosslinking ratio at a fixed total crosslinking density, the amplitude of the stress-relaxation can be controlled. Moreover, it is also possible to change the stress-relaxation timescales of the hydrogels using different metal ions [43], making the SRG hydrogel an excellent platform to study the cell responses to the stress-relaxation of ECM.

## 3. Materials and Methods

### 3.1. Materials

Acrylate-terminated 4-armed polyethylene glycol (4-Armed PEG-Aclt, Mw: 20 kDa) was purchased from JenKem, Inc., Beijing, China. Acrylate-connected histidine-rich peptides (acrylate-Gly-His-His-Pro-His-Gly-His-His-Pro-His, named acrylate-PH_6_ hereafter) and thiol-connected RGD peptide (cyclo(-Arg-Gly-Asp-D-Phe-Cys)) were purchased from GL Biochem (Shanghai, China) Ltd. Unless otherwise stated, all other reagents were purchased from Sinopharm Chemical Reagent Co., Ltd. (Beijing, China). NCTC clone 929 (L cell, L-929, derivative of Strain L), HeLa, and MCFs were provided by the Cell Bank of the Chinese Academy of Sciences (Chinese Academy of Sciences, Shanghai, China). 

### 3.2. Correlation between Young’s Modulus and Immersion Time

Acrylate-PH_6_-peptides, acrylamide, and 4-armed PEG-Aclt were dissolved in Milli-Q water at 60 mg mL^−1^, 225 mg mL^−1^, and 60 mg mL^−1^, respectively. The solution was degassed under the protection of argon and sonicated for 3 × 5 min to remove the dissolved oxygen. Lithium phenyl-2,4,6-trimethylbenzoylphosphinate (LAP) was added to all the solutions as the photoinitiator at a concentration of 0.5 mg mL^−1^. Hydrogel precursors were injected into band-shaped molds before polymerization under UV illumination (285 nm, 8 W) for 2 h at room temperature. Transparent hydrogels were obtained and extensively dialyzed in deionized water to remove the unreacted monomers. 

Then the hydrogels were cut into 20 pieces and immersed in Tris buffer solution (1 M Tris and 0.3 M KCl, pH = 7.6) containing 2.2 mM ZnCl_2_. The Young’s modulus of these hydrogels was determined every 15 min for the next 9 h (Appendix A). The Young’s modulus (*E*) of the hydrogel after being immersed for different times (*t*) can be summarized as *E* = 19.62*t* + 197.73.

### 3.3. Preparation of the SRG Hydrogel

Acrylate-PH_6_-peptides, acrylamide, and 4-armed PEG-Aclt were dissolved in Milli-Q water and mixed to prepare the precursors. Three kinds of hydrogel precursors at different concentrations of 4-armed PEG-Aclt (20, 40, and 60 mg mL^−1^) were prepared, while the concentrations of acrylate-PH_6_ peptides and acrylamide remained at 60 and 225 mg mL^−1^. All the solutions were degassed under the protection of argon and sonicated for 3 × 5 min to remove the dissolved oxygen. LAP was added to all the solutions as the photoinitiator at a concentration of 0.5 mg mL^−1^. Then, the three kinds of hydrogel precursors were injected into band-shaped molds layer by layer before polymerization under UV illumination (285 nm, 8 W) for 2 h at room temperature. Transparent hydrogels (width: 5.8 mm, length: 73 mm, thickness: 1.23 mm) were obtained and extensively dialyzed in deionized water to remove the unreacted monomers. 

Then, the band-like hydrogel was held by a clamp and gradually descended into Tris buffer solution (1 M Tris and 0.3 M KCl, pH = 7.6) containing 2.2 mM ZnCl_2_ to trigger the formation of gradient coordination. The descent speed was kept at ~0.166 mm min^−1^ according to the correlation between Young’s modulus and the immersion time. For the hydrogels used for cell culture, acrylate-terminated RGD peptide was added to the precursor to a concentration of 1.0 mg mL^−1^, and the hydrogels were prepared as described above.

### 3.4. Scanning Electron Microscopy (SEM)

Scanning electron microscopy (SEM) images were obtained using a Quanta scanning electron microscope (Quanta 200, FEI, Hillsboro, OR, USA) at 20 kV. The hydrogels were lyophilized prior to the measurement. Then, all the samples were sputter-coated with platinum and imaged.

### 3.5. Mechanical Test

The mechanical properties of the hydrogel were determined using a tensile-compressive tester (Instron-5944 with a 10 N sensor) in the air at room temperature, and the illustration is shown in Appendix A. For the compression-crack test, the compression rate was maintained at 0.33 mm min^−1^ to 0.35 mm min^−1^ (~20% original thickness per minute) depending on the original thickness of the hydrogel. For the compression-relaxation cycle test, the compression rate was also maintained at 0.33 to 0.35 mm min^−1^ (~20% original thickness per minute) depending on the original thickness of the hydrogel, and the compression strain was maintained at 0.8 to 0.9 mm min^−1^ (~50% original thickness) depending on the original thickness of the hydrogel. For the stress-relaxation experiments, a compressive strain of 50% was applied to the hydrogels quickly (in ~0.2 s), and then the relaxation of the stress was recorded. The characteristic relaxation times were determined by fitting the stress-relaxation curves with an exponential model.

### 3.6. Swelling Ratios and Water Contents

Five typical locations on band-shaped hydrogel were cut, and their original volumes were recorded as *V*_1_. Then the hydrogels swelled in deionized water for 24 h and immersed in Tris buffer solution (1 M Tris and 0.3 M KCl, pH = 7.6) containing 2.2 mM ZnCl_2_ for different immersion times. Then, the volume of the hydrogels was recorded as *V*_2_. The swelling ratios of typical locations on the SRG hydrogel were equal to *V*_2_/*V*_1_.

The dry weights of hydrogels after swelling were recorded as *W*_1_. Then the hydrogels were lyophilized, and the weights were recorded as *W*_2_. The water content of the hydrogels was equal to (*W*_1_ − *W*_2_)/*W*_1_.

### 3.7. Cell Culture

L929, HeLa, and MCFs cells were cultured in 89% standard Dulbecco’s modified Eagle’s medium (DMEM, Gibco, Grand Island, NY, USA). Then, 10% bovine serum (Gibco, NY, USA) and 1% penicillin/streptomycin (Gibco, NY, USA) were added to the medium. The medium was changed every 3~4 days.

For the cell culture on hydrogel, L929, HeLa, and MCFs cells were trypsinized using 0.05% trypsin (Invitrogen) and washed using serum-free DMEM. Then, the cells were seeded on the hydrogels at a density of 10^5^ mL^−1^. The incubation of cells was performed at 37 °C and 5% CO_2_. After 24 h, cell viabilities were evaluated using the Cell Counting Kit-8, following the manufacturer’s instructions (Dojindo Laboratories, Shanghai, China).

### 3.8. Immunostaining and Cell Morphology Analysis

The L929 cells culture medium was first removed from the hydrogels. Then, L929 cells on the hydrogel were fixed with 4% paraformaldehyde at 37 °C for 10 min. L929 cells were then washed three times in PBS containing calcium (cPBS) and incubated for 30 min in 1% Triton. The samples were stained with 0.1% phalloidin (Invitrogen, Carlsbad, CA, USA) for 90 min and DAPI for 1 min (Invitrogen, Carlsbad, CA, USA). Finally, images were obtained using an OLYMPUS-IX73 fluorescence microscope (OLYMPUS, Tokyo, Japan). Cell morphology analysis was performed using ImageJ.

Statistical significance was determined using the Student’s *t*-test. Statistical significance was set to a *p*-value < 0.05.

## 4. Conclusions

In conclusion, we developed a hybrid network hydrogel with a constant Young’s modulus and controllable stress-relaxation gradient by altering the coordinate and covalent crosslinker ratios. The gradual stress-relaxation of hydrogels along the coordination crosslinking ratios was studied in detail. Based on the hydrogel, the effects of the amplitude of the stress-relaxation on cell spreading behaviors were investigated. The study in this work indicates that cell spreading is suppressed at an increasing stress-relaxation amplitude with a fixed elasticity and stress-relaxation timescale. We anticipate that the hydrogel can find broad applications in systematic investigations of the stress-relaxation gradient on cell behaviors. The design principle in this study also represents a general route to spatially tune the stress-relaxation of hydrogels without affecting their components and elasticity.

## Figures and Tables

**Figure 1 ijms-23-05170-f001:**
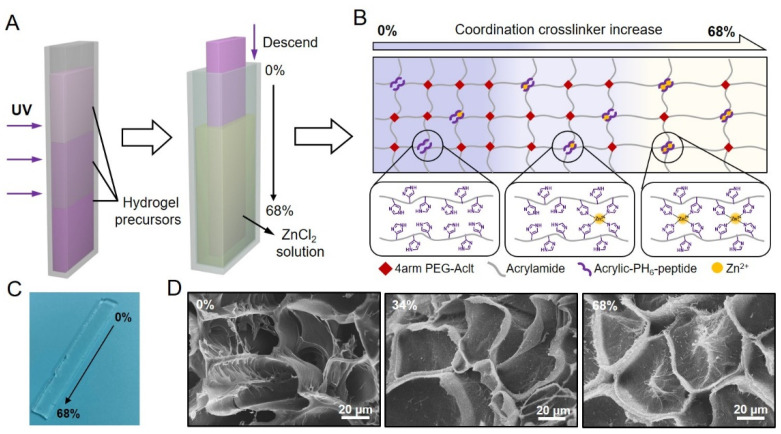
Design, preparation and microstructure of the SRG hydrogel. (**A**) Preparation of the hydrogel with stress-relaxation gradients. The precursor solutions were injected into the mold layer by layer and gelated under UV illumination. Then the as-prepared hydrogel was held by a clamp and gradually descended into the ZnCl_2_ solution to achieve gradient coordination. (**B**) Schematic diagram of the hydrogel network with stress-relaxation gradients. The PH_6_-Zn^2+^ crosslinker ratio increases from 0% to 68% (left to right). (**C**,**D**) Optical image of the SRG hydrogel (**C**) and SEM images corresponding to the microstructures of three typical locations (0%, 34%, and 68%) of the SRG hydrogel (**D**).

**Figure 2 ijms-23-05170-f002:**
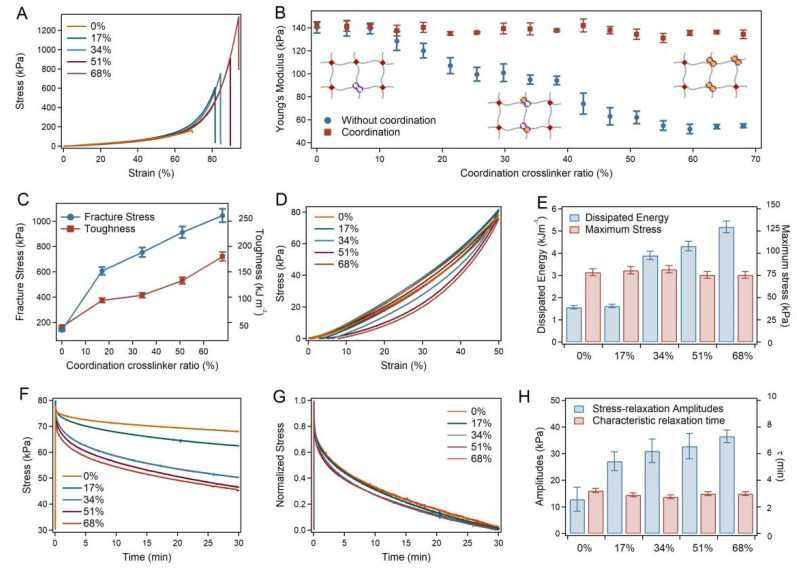
Mechanical properties of SRG hydrogels. (**A**) Typical stress−strain curves of the SRG hydrogels. (**B**) Young’s modulus of different locations on the SRG hydrogels in the presence and absence of Zn^2+^. Blue points are the Young’s modulus of hydrogels without coordination at the same locations corresponding to those of hydrogels with coordination. (**C**) Summarized fracture stress and toughness corresponding to different locations on the SRG hydrogels. (**D**) Compression−relaxation cycles of different locations on the SRG hydrogel. (**E**) Summarized maximum stress and dissipated energy corresponding to different locations on the SRG hydrogels. (**F**,**G**) Stress−relaxation curves (**F**) and normalized stress−relaxation curves (**G**) of different locations on the hydrogel under compressions for 30 min at a stain of 50%. (**H**) Summarized stress−relaxation amplitudes and characteristic relaxation time (*τ*) at different locations of SRG hydrogels.

**Figure 3 ijms-23-05170-f003:**
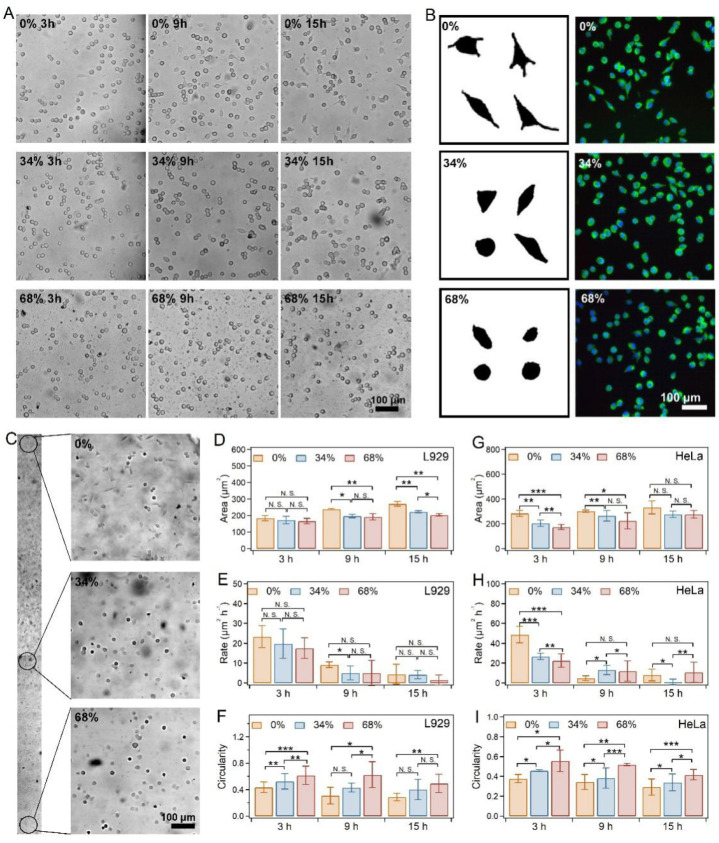
Cells spreading on SRG hydrogels. (**A**) Representative images of L929 cells on different locations of the SRG hydrogels after being cultured for different times. (**B**) Representative profiles and immunostaining of L929 cells on different locations of the SRG hydrogels. The nuclei and F−actin of cells were stained using DAPI (blue) and phalloidin (green), respectively. (**C**) Images of L929 cells spread on different locations of the SRG hydrogels. The image was combined using 18 images taken with a microscopy. (**D**–**F**) Spreading area (**D**), spreading rate (**E**), and circularity (**F**) of L929 cells on SRG hydrogels. (**G**–**I**) Spreading area (**G**), spreading rate (**H**), and circularity (**I**) of HeLa cells on SRG hydrogels. Asterisks denote statistical significance followed by *t*−test (*p* > 0.05: N.S.; *p* < 0.05: *; *p* < 0.01: **; *p* < 0.001: ***).

**Figure 4 ijms-23-05170-f004:**
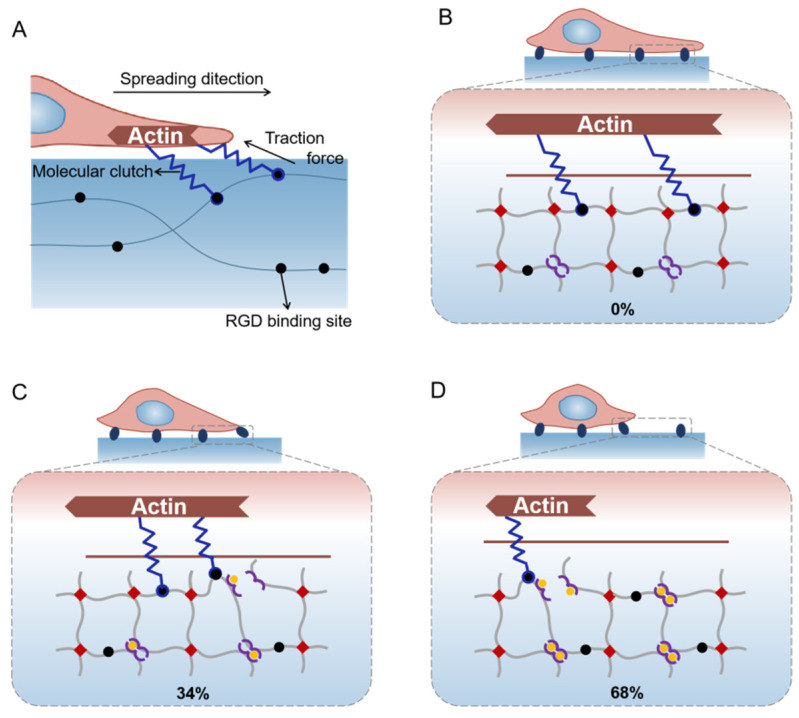
Hypothesis of mechanisms for cell spreading regulation with gradual stress-relaxation. (**A**) Schematic of cell spreading on ECM. ECM (extracellular matrix) coupled clutches inhibit retrograde flow of actin and drive cell spreading. (**B**–**D**) Cells spreading on different locations of the SRG hydrogels (0%, 34% and 68%).

## Data Availability

All data are available in the main text or the Appendix A.

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
