# Peer review of "Gradual Stress-Relaxation of Hydrogel Regulates Cell Spreading"

_ijms, 2022, doi:10.3390/ijms23095170_

Round 1

Reviewer 1 Report

The work by Yu et al. reports the preparation of a hybrid hydrogel with a constant Young’s modulus and a stress-relaxation gradient that can be spatially controlled by altering the covalent and coordination crosslinking ratios. The amplitude of stress-relaxation gradient influences the spreading of three different cell lines. Overall, the manuscript is well written and very significant to the field; the results may open up the way for further research.

Here are some comments about the work that should be addressed before its publication.

  1. Figure 1B well represents the physical crosslinking but in my opinion should be made clearer. In Figure 1A the gel is represented vertical, while in 1B it’s horizontal. To make this picture clearer, I would turn 1B vertical, or I would indicate the axis somewhere in both 1A and 1B or insert an axis with the percentage 0% to 68% in figure 1A.
  2. Some citations of recent works dealing with this topic should be added in the introduction, such as Biomater. Sci., 2022,10, 270-280. In particular, the authors should add a comment/hypothesis on why the fundings of this previous research (made on fibroblast) state the opposite of their current work, i.e. that fibroblasts elongate, migrate and proliferate better in hydrogels that display a higher stress relaxation amplitude.
  3. Figure 2B. What does the x-axis represent for the blue points (gel without coordination) since it is not the % of coordination crosslinker? This should be clearly stated in the paper.
  4. Figure S2. Correct the first sentence: “The difference of Young's modulus vs(?) immersion time: (A) Typical stress-strain curves of hydrogels after being immersed ….
  5. The experimental part of swelling and water content measurements (Figure S4) should be described in the Materials and Methods section.

Reviewer 2 Report

In the manuscript “Gradual stress-relaxation of hydrogel regulates cell spreading” by Yu et al., authors develop an hydrogel with a controllable stress-relaxation gradient and a constant elasticity for the study of cell behavior. The research sounds very interesting, but some revisions must be addressed before a possible publication in this journal.

The raised concerns are listed below:

  • In the introduction, line 46, authors stated “on the other hand…” In my opinion, it is not clear if they want to talk about something different from cell behaviour, topic already introduced in the first paragraph. Please verify the organization of the first two paragraphs.
  • Introduction, line 54-55: “effects ON cell behavior”
  • Introduction, line 56-59: please add some references that support the definition of ratio of crosslinking, intrinsic kinetics and overall density.
  • Materials and Methods: please indicate the origin and type of each cell line used, because only for L929 it is indicated that they are mouse fibroblasts.
  • Materials and Methods, line 138: which is the cell density used?
  • Have you performed any statistical analysis on the presented data? Please, check your data and add the statistical investigation in the text and in the graphs.
  • Figure 3, panels D – I: label “L929” and “HeLa” could be helpful for the reader to identify the results obtained on the two different cell line.
  • I would recommend the authors to stress about the findings of the work in the conclusion, specifying the application they have selected for the study.
